# AI-Aided Design of Novel Targeted Covalent Inhibitors against SARS-CoV-2

**DOI:** 10.3390/biom12060746

**Published:** 2022-05-25

**Authors:** Bowen Tang, Fengming He, Dongpeng Liu, Fei He, Tong Wu, Meijuan Fang, Zhangming Niu, Zhen Wu, Dong Xu

**Affiliations:** 1Department of Electrical Engineering and Computer Science, Informatics Institute, Christopher S. Bond Life Sciences Center, University of Missouri, Columbia, MO 65211, USA; bowen@mindrank.ai (B.T.); dpliu@uchicago.edu (D.L.); hef740@nenu.edu.cn (F.H.); tongywu@gmail.com (T.W.); 2Fujian Provincial Key Laboratory of Innovative Drug Target Research, School of Pharmaceutical Sciences, Xiamen University, Xiamen 361000, China; fengming_he@outlook.com (F.H.); fangmj@xmu.edu.cn (M.F.); 3MindRank AI Ltd., Hangzhou 310000, China; zhangming@mindrank.ai; 4School of Information Science and Technology, Northeast Normal University, Changchun 130117, China; 5Department of Epidemiology and Statistics, Institute of Basic Medical Sciences, Chinese Academy of Medical Sciences, School of Basic Medicine, Peking Union Medical College, Beijing 100006, China

**Keywords:** COVID, SARS-CoV-2, 3C-like main protease, drug design, deep Q-learning network

## Abstract

The drug repurposing of known approved drugs (e.g., lopinavir/ritonavir) has failed to treat SARS-CoV-2-infected patients. Therefore, it is important to generate new chemical entities against this virus. As a critical enzyme in the lifecycle of the coronavirus, the 3C-like main protease (3CLpro or Mpro) is the most attractive target for antiviral drug design. Based on a recently solved structure (PDB ID: 6LU7), we developed a novel advanced deep Q-learning network with a fragment-based drug design (ADQN–FBDD) for generating potential lead compounds targeting SARS-CoV-2 3CLpro. We obtained a series of derivatives from the lead compounds based on our structure-based optimization policy (SBOP). All of the 47 lead compounds obtained directly with our AI model and related derivatives based on the SBOP are accessible in our molecular library. These compounds can be used as potential candidates by researchers to develop drugs against SARS-CoV-2.

## 1. Introduction

The emerging coronavirus SARS-CoV-2 has caused an outbreak of coronavirus disease 2019 (COVID-19) worldwide [1]. By the end of May 2022, the Johns Hopkins Coronavirus map tracker had reported more than 527 million SARS-CoV-2 infections and over six million deaths [2]. The number of infections and deaths is still increasing. To deal with the threat of SARS-CoV-2, it is necessary to develop new inhibitors or drugs. Unfortunately, since the outbreak of severe acute respiratory syndrome (SARS) 18 years ago, there has been no approved treatment for SARS-associated coronavirus (SARS-CoV) [3], which is similar to SARS-CoV-2. SARS-CoV-2-injected patients have also failed to respond to repurposed drugs, such as lopinavir and ritonavir [4]. Structure-based antiviral drug design with a novel artificial intelligence algorithm may be an effective approach to developing SARS-CoV-2-targeted inhibitors or drugs. Owing to the efforts of many researchers, several pieces of important information about the viral genome and protein structures are currently available. We know that non-structural protein 5 (Nsp5), a cysteine protease, is one of the main proteases (M^pro^) of SARS-CoV-2, also known as “3C-like protease” (3CL^pro^). Moreover, we know that the 3D structure of 3CL^pro^ is very similar to that of SARS-CoV with a sequence identity of >96% and 3D structure superposition RMSD_Cα_ of 0.44 Å, as shown in Appendix A.

Overall, 3CL^pro^ has been reported as an attractive target for developing anti-coronavirus drugs for the following reasons: (1) this protease is highly conserved in both sequences and 3D structures [5], (2) 3CL^pro^ is a crucial enzyme for the replication of related viruses (including SARS and SARS-CoV-2), and (3) it only exists in viruses and not in humans. Developing specific antiviral drugs that target the 3CL^pro^ of viruses has shown significant success; for example, both lopinavir and ritonavir (approved HIV drugs) can completely occupy the substrate-binding site of 3CL^pro^, thereby disrupting the replication of human immunodeficiency virus (HIV). However, due to the large difference between HIV and SARS-CoV-2 3CL^pro^, lopinavir and ritonavir are ineffective for inhibiting SARS-CoV-2 [4]. On the other hand, the substrate-binding site of 3CL^pro^ is almost the same between SARS-CoV-2 and SARS, as shown in Appendix A. Therefore, the developed potential inhibitors and drug design experience for SARS-3CL^pro^ may also apply to SARS-CoV-2. For example, the recently solved structure of SARS-CoV-2 3CL^pro^ (PDB ID: 6LU7) demonstrated that the developed inhibitor N3 [6], which is a covalent inhibitor derived from non-covalent inhibitors against SARS, can also bind to SARS-CoV-2 3CL^pro^ with a similar binding conformation (Appendix A).

The available information from previous research can aid the design of novel targeted covalent inhibitors (TCIs) [7] against SARS-CoV-2. A successful TCI against 3CL^pro^ must first be able to fit in the binding site of 3CL^pro^ with an appropriate pose that keeps its reactive groups close enough to Cys145, which then undergoes a chemical step (nucleophilic attack by Cys145), leading to the formation of a stable covalent bond, as presented in the scheme below:$$3{\mathrm{CL}}^{\mathrm{pro}}-145\mathrm{CYS}+\mathrm{TCI}\;\underset{k_{2}}{\overset{k_{1}}{⇄}}\;3{\mathrm{CL}}^{\mathrm{pro}}-145\mathrm{CYS}\cdot \mathrm{TCI}\underset{k_{4}}{\overset{k_{3}}{⇄}}\;3{\mathrm{CL}}^{\mathrm{pro}}-145\mathrm{CYS}-\mathrm{TCI}$$

In theory, TCIs usually have a longer target residence time compared with that of non-covalent inhibitors, given the following: (1) for inhibition, k_1_ must be larger than k_2_, and non-covalent binding is determined by the equilibrium constant k_1_/k_2_; (2) TCIs undergo a chemical reaction with the target, where usually k_3_ >> k_4_; and (3) for TCIs, the binding process is controlled by k_1_ k_3_/(k_2_ k_4_), which is larger than k_1_/k_2_ for non-covalent inhibitors. In some extreme cases, k_4_ = 0; thus, these inhibitors covalently bind until the target is no longer detectable [7,8].

Considering that the inhibitors of SARS 3CL^pro^ may exert bioactivity against SARS-CoV-2, we have created a molecular library that contains all of the reported SARS-3CL^pro^ inhibitors (284 molecules) [9,10,11,12,13,14,15,16,17,18,19,20,21,22,23,24,25,26,27,28,29]. We will also add new validated molecular structures to this library as our research progresses. To date, there are no clinically approved vaccines or drugs specifically targeting SARS-CoV-2. Therefore, to discover novel candidate drugs targeting SARS-CoV-2, we combine artificial intelligence (AI) with fragment-based drug design (FBDD) to accelerate the generation of potential lead compounds and design TCIs.

AI, especially deep learning, has been used to predict molecular properties [30,31,32,33] and design novel molecules [34]. Table 1 lists some deep-learning-based tools and their features. Among these tools, the tool involving recurrent neural networks relies on a large number of training molecules with known structures to learn the context information as the basis for molecule generation. Over time, various advanced generative tools were proposed, including GENTRL, ORGAN, and ORGANIC. These tools were expected to increase success rates by introducing posterior distributions and adversarial attacks as regularizers. However, using the generative models still does not guarantee that their outputs are valid molecules for drug discovery; to overcome such problems, Molecule Deep Q-Networks (MolDQNs) with reinforcement learning (RL) were proposed. MolDQN can iteratively work at the atomic level to generate molecules fitting all predefined constraints. Hence, all outputs are chemistry-reasonable molecules.

Considering the advantages of MolDQN, we combined the framework with more fragment-based rules to optimize the generated molecules in this study. In contrast to earlier deep-learning molecular design by adding a single atom one at a time [30,34], our approach explores new molecules by adding a meaningful molecular fragment one by one, which is computationally more efficient and chemically more reasonable. For our AI model, we initially prepared the molecular fragment library by using the collected SARS-CoV 3CL^pro^ inhibitors (284 molecules) targeting SARS-CoV-2 3CL^pro^, as shown in Figure 1. Then we split this set of molecules into fragments with a molecule weight of no more than 200 Da. Both of the collected inhibitors and the fragments can be found at [40]. Then we applied an advanced deep Q-learning network with fragment-based drug design (ADQN–FBDD) to generate potential lead compounds. If researchers have sufficient experience and internal lead compounds or biased fragments, they can manually add the lead compounds and biased fragments to the corresponding files of the ADQN–FBDD. Using the same fragments directly from existing bioactivate molecules, our ADQN–FBDD agent could easily access the potential chemical space for the 3CL^pro^ of SARS-CoV-2.

After the ADQN–FBDD automatically generated novel compounds targeting 3CL^pro^, we obtained a covalent lead compound library with 4922 unique valid structures. A total of 47 of these compounds were selected with high scores with our AI model’s reward function. These molecules were further evaluated through docking studies. Among the 47 lead compounds, compound #**46**, with a high covalent docking score, attracted our attention, showing a small difference between the non-covalent and covalent docking poses. After carefully examining the interaction mode of lead #**46** with 3CL^pro^, we believe that there is still much space for optimization. Subsequently, we designed a series of derivatives from compound #**46** based on our chemical biology knowledge and the structure-based optimization policy. All of the generated molecular structures are published in our code library, https://github.com/tbwxmu/2019-nCov [40]. We would encourage researchers interested in finding a potential treatment for COVID-19 to synthesize and evaluate some of these molecules.

## 2. Results

Integrating the double-dueling deep-Q-learning network with fixed q-targets and prioritized experience replay allowed our agent to be stable and efficient when learning from the chemical environment. With a combination of the state-of-the-art AI algorithm and FBDD, as shown in Figure 2, the approach was flexible and efficient in accessing the focused chemical space for SARS-CoV-2 3CL^pro^. Based on the configurations targeting SARS-CoV-2 3CL^pro^, the ADQN–FBDD generated a lead library containing 4922 unique molecular structures (Appendix A dataset in [40]). To narrow our focus to a smaller set of molecules for analysis, we defined the filter rules (QED > 0.1 and DRL-reward ≥0.6); detailed information on the rules can be found in the Methods section. A total of 47 unique molecules (Appendix A) were selected for non-covalent docking and covalent docking evaluation. These 47 virtual leads exhibited an appropriate 3D complexity with typical characteristics of peptidomimetics and protein–protein interaction (PPI) inhibitors. They were mainly ranked according to the covalent docking scores, considering that covalent docking scores also include the scores of non-covalent docking [41].

To analyze the common features of these generated molecules, we used the Canvas Similarity and Clustering tool. As shown in Appendix A, the 47 leads were clustered into five clusters according to the clustering metric (shape). Among the five clusters, all of the molecules in Cluster #1 contained R-shaped pyrrolidin-2-one structures and may be considered optimal segments of the S2 subsite or S1 subsite. In addition, they contained some substituents with hydrophobic cyclic groups, such as aromatic rings, alicyclic rings, and aliphatic hydrocarbons. The molecules in Cluster #2 included 2-pyrrolidone in their structures, including R-shaped, S-shaped, and S’-shaped. Furthermore, their side chains linked to covalent targets contained large substituents (e.g., aromatic rings) and small substituents (e.g., aliphatic hydrocarbons or aliphatic amines). Cluster #3 comprised several straight chain-like molecules with substituted or non-substituted pyrrolidin-2-one, which may also be regarded as optimal segments of the S2 subsite or S1 subsite. In addition, their α-oxo aldehyde group can form a covalent bond with Cys145. Cluster #4 mainly consisted of 2-pyrrolidone, heterocyclic rings, and heteroaromatic rings, whose polarities were considerably higher than those of the molecules from other clusters, which may reduce the affinities of the protein active pockets S1 and S2. Their covalent targets included α, β-unsaturated ketones. Each molecule in Cluster #5 had four extension directions with different covalent targets, matching four sub-pockets of active protein-binding sites. This could contribute to a high binding affinity; hence, molecules in Cluster #5 were the best candidates for further analysis. We also considered the RMSD difference between the covalent and non-covalent binding poses based on all heavy atoms. Finally, we selected lead molecule #**46** (Figure 3), with a good covalent docking score and a small RMSD value (docking affinity: −8.722 kcal/mol, KabschRmsd 1.71 Å) in Appendix A. The FBDD approach was further optimized to obtain a series of derivatives.

Molecule #**46** was ranked first based on the covalent docking score, and its interaction model with the binding site was carefully examined, as shown in Figure 3. Although compound #**46** had the best covalent docking score, it had an aldehyde group. Interestingly, upon a detailed examination of #**46**, it was found that its interactions were similar to those of the inhibitor 6XHO (PF-00835231) designed by Pfizer, thus demonstrating the potential of our proposed method. Moreover, 6XHO was reported to have modest levels of irreversible inhibition, which allowed co-crystallization in complex with SARS-CoV-2 3CL [42]. Based on PF-00835231, Pfizer further developed Lufotrelvir (PF-07304814). The phosphate group of PF-07304814 contributed to improved solubility and was cleaved in vitro to release the active antiviral PF-00835231. Similar to remdesivir, PF-07304814 was administered via intravenous infusion. Moreover, Pfizer developed the orally active 3C-like protease inhibitor Nirmatrelvir (PF-07321332), crystallized as 7VH8. We superimposed lead #**46** with 6XHO and 7VH8 and compared their 3D pose and interactions, as shown in Figure 4. Most of their binding sites were aligned with their SARS-CoV-2 3CL^pro^ ligands. Considering that there is still much space for compound #**46** to fill in the S1′ subsite and that α-ketoamides may be suitable to fit into the oxyanion hole (Figure 5A) of 3CL^pro^ [3], we replaced the aldehyde with formamide and also replaced the 1,4 Michael acceptors with α-ketoamides. Therefore, we optimized compound #**46** to obtain compound **46–14–1** (Figure 5A,B).

The non-bonding interactions between compound **46–14–1** and SARS-CoV-2 3CL^pro^ are mainly hydrogen bonds (H-bonds, five in total). The carbonyl group of the covalent scaffold α-ketoamide forms H-bonds with Leu141 and Gly143 as a hydrogen acceptor or a hydrogen donor. The hydrogen on the nitrogen atom of the triazole ring forms an H-bond with Glu166, and Glu166 forms an H-bond with the carbonyl group on the main chain. The oxygen of the β-lactam ring forms an H-bond with His41. To enhance the polarity of the compounds, sulfonic groups were introduced to replace the ketone carbonyl groups on the main chain, and sulfonamides **46–14–2** were obtained. The covalent docking model of compound **46–14–2** with SARS-CoV-2 3CL^pro^ is shown in Figure 5C,D.

To make compound **46–14–2** fit the active pocket with a higher affinity, we added a carbon atom to the sulfonic acid group of the original molecule, and it extended the carbon chain to increase the molecule’s flexibility and obtained another optimized compound **46–14–3**. The mode of covalent docking with SARS-CoV-2 3CL^pro^ is shown in Figure 6. Due to the introduction of carbon atoms and the enhancement of molecular flexibility, the β-lactam ring can be inserted deeper into the S2 pocket, and other fragments of the compound can better adapt to the S1, S1′, and S3 subsites. The α-carbonyl carbon on the α-ketoamide of compound **46–14–3** forms a covalent bond with the key residue Cys145 on the protease; however, the primary non-bonding interaction is still an H-bond (indicated by a yellow dash). The triazole ring mainly forms H-bonds with the Phe140 and Glu166 residues in the S1 pocket. The α-ketoamide covalent binding fragment mainly forms H-bonds with the key amino acid residues Cys145, Gly143, and Ser144 in the S1′ subsite, forming an oxyanion hole in the red circle shown in Figure 5. The β-lactam side chain mainly forms H-bonds with residues Tyr54 and AsS187 in the S2 pocket. The chromone scaffold mainly forms H-bonds with the key residues Thr190 and Gln192 in the S3 pocket. In addition, the oxygen on the sulfonyl group of the main chain forms an H-bond with Glu166 in the S1 pocket.

The optimized compounds (**46–14–1**, **46–14–2**, and **46–14–3**) are shown in Figure 7 that may be further evaluated by molecular dynamics simulation to determine the binding free energy and by quantum chemical calculation to determine the reaction energy barrier. Furthermore, **46–14–1**, **46–14–2**, and **46–14–3** were chosen as candidates for chemical synthesis and anti-SARS-CoV-2 activity testing, which is ongoing.

## 3. Discussion

Computational approaches are particularly important for emerging diseases, given the need to provide timely solutions. In this study, our robust and efficient computational method and pipeline for designing compounds can provide useful drug candidates for treating SARS-CoV-2 infections. More information about our AI model-generated leads and derivatives can be found at [40]. These candidates or their variants have a high probability of producing valid leads of anti-COVID-19 drugs. Nevertheless, the computational design requires experimental validation. Although we are currently conducting experimental validations, we hope that other researchers could use these candidates to accelerate the development of anti-COVID-19 drugs, given the urgency of treating the disease.

FBDD has become a key technology in the pharmaceutical industry for early stage drug discovery and development in the last two decades. Molecular fragments are always small in size compared with intact molecules. Several fragments are especially favorable to some protein binding pockets. For example, the α-ketoamides can form a covalent bond with CYS145 of 3CL^pro^ in SARS-CoV-2 and fit into the oxyanion hole, as Figure 6B presented. These fragments with high-quality interactions with protein targets can more likely grow up or integrate into lead compounds. During this growing or integrating process, ADMET (absorption, distribution, metabolism, excretion, and toxicity) properties could be easier to control than working directly on the intact molecule. Based on the above advantages of FBDD, we believe more and more machine-learning- or deep-learning-based molecular generative models will use FBDD to design molecules in the future.

Compared with other deep RL methods, the ADQN–FBDD has several notable features: (1) It directly modifies and generates molecular structures without format conversion problems; other tools, such as in silico Medicine’s GENTRL (https://insilico.com), may generate invalid SMILES. (2) Most generative models require pretraining on a specific dataset and produce molecules with high similarities to a given training set. For example, the best molecule from Insilico Medicine for the target DDR1 is highly similar to the kinase inhibitor Iclusig (ponatinib) [43]. The ADQN–FBDD does not require any pre-training and is capable of generating new molecules. (3) The process of generating molecules is highly efficient, as the ADQN–FBBD is a molecular-fragment-based model with knowledge of chemical reactions, whereas other models are all atom-based, with no rules for chemical reactions [30,38,44]. (4) The ADQN–FBDD is highly flexible and user-friendly for medicinal chemists, who can easily introduce their drug discovery experience into the reward function to guide novel molecule generation. Our ADQN–FBDD and related pipeline may be used not only for designing anti-COVID-19 drugs but also for other structure-based drug discoveries, especially for emerging infectious diseases that require timely treatment.

## 4. Methods

### 4.1. Markov Decision Process (MDP) for Molecule Generation

Intuitively, the problem of chemical structure graph generation can be formulated as learning a reinforced agent, which performs discrete actions of chemical-reaction-based fragment addition or removal in a chemistry-aware MDP. Formally, the components of the MDP include *M* = {*S*, *A*, *P*, *R*, *γ*}, where each term can be defined as follows:

$S=\left\{S_{t}\right\}$ denotes the state space containing all possible intermediate and final generated molecular graphs. Each $s_{t}$ is a tuple of (*s*, *t*); *s* represents a valid molecular structure, and *t* is the time step. For the initial state, $s_{0}$, the structure may be represented as a specific core, as demonstrated in Figure 2C, or randomly chosen from the prepared fragment library at time *t* = 0. We limited the maximum number of time steps, *T*, in the fragment-based MDP, which defines the set of terminal states as {$s_{t}$|*t*=*T*}, containing all the states, with the number of steps reaching the maximum allowed value *T.*

Ac *=* {*A_t_*} denotes a set of actions that describe the modification made on the current molecular structure at each time step, *t.* Each action can be classified into three categories: fragment addition, fragment deletion, and no modification.

*P = p* (*s_t_*_+1_*|s_t_ … s*_0_) *= p (s_t+1_|s_t_*, *a_t_)* is the basic assumption in the MDP. The state transition probability specifies the next possible state given the current state and action at time step, *t*. Here, we defined the state transition to be deterministic. For example, for *S*_0_ to *S*_1_ (Figure 2C), by adding a 1*H*-1,2,3-triazol-4-yl fragment on *S*_0_, the next state *S_1_* would be the new structure consisting of added 1*H*-1,2,3-triazol-4-yl with a probability of 1.

*R* is the reward function that specifies the reward after reaching state $S_{t,}\;\mathrm{and}\;\gamma \epsilon \left(0,1\right]$ is the discount factor; typically, γ = 0.9 in our study. In our framework, the state always had a valid and complete chemical structure at each step, as shown in Figure 2C. A reward was given not just at the terminal states but also at each step. Both intermediate rewards and final rewards were used to guide the behavior of the RL agent. Therefore, there was no delayed or sparse reward issue as experienced with many other reinforced frameworks [36,45]. Additionally, to ensure that the last state was rewarded the most, we used *γ^T^*^–*t*^ to discount the value of the rewards at state $s_{t}$. Our reward function can directly integrate the experience of medicinal chemists. For example, given a core of interest, medicinal chemists may add biased fragments of interest. Their input can be used to design a reward function that gives high reward signal values to biased fragments so that the ADQN–FBDD may have a better chance of generating the desired structures.

### 4.2. Chemical Environment Design

In our RL framework, the chemical environment receives action, *a*_t_, from the agent and emits scalar reward, r_t_, and state $s_{t}^{\prime }$ to the agent, as shown in Figure 2A. Notably, the definition of the environment state differed from the general approach such that the environment state consisted of only the environment’s private representation invisible to the agent. We defined the state of the chemical environment, $s_{t}^{\prime }$, as the generated intermediate molecular structure at time step, *t*, as is fully observable by the RL agent. Basically, the environment’s state, $s_{t}^{\prime }$, was equivalent to the agent’s state, $s_{t+1}.$ For the task of molecule generation, the environment can incorporate the rules of chemistry. In our study, chemistry rules were not only about chemical valency but also about adding and removing the fragments of known inhibitors derived from chemical reactions. The detailed information of 45 defined chemical reaction rules is presented in Appendix A.

### 4.3. Agent Design

As shown in Figure 2A, the basic model of our ADQN–FBDD was an advanced Q-network. The goal of molecule generation was equally to fit a Q function *Q*(*s_t_*, *a_t_*) to make the agent choose the action *a_t_* at state *s_t_* to maximize the expected γ-discounted cumulative rewards with policy π. Mathematically, given the agent’s policy π, the value of the state-action pair *Q*^π^(*s_t_*, *a_t_*) and the value of state *V*^π^(*s_t_* ) can each be defined as follows:(1)$$Q^{\pi \;}\left(s_{t},a_{t}\;\right)=E_{\;a_{t}∼\pi \left(s_{t}\right)}\left[\overset{T}{\underset{n=t}{\sum }}\gamma^{T-n}\cdot R\left(s_{n},\;a_{n}\right)\right]=E_{\;a_{t}∼\pi \left(s_{t}\right)}\left[R\left(s_{t},\;a_{t}\right)+\gamma \cdot E_{\;a_{t+1}∼\pi \left(s_{t+1}\right)}(Q^{\pi \;}\left(s_{t+1},a_{t+1}\;\right)\right]$$
(2)$$\;\;\;\;\;\;\;V^{\pi \;}\left(s_{t}\right)=E_{a_{t}∼\pi \left(s_{t}\right)}\left[Q^{\pi \;}\left(s_{t},a_{t}\;\right)\right]\;\;\;\;$$
where $E_{\;a_{t}∼\pi \left(s_{t}\right)}$ is the expectation within policy π on state *s_t_* with *a_t_*, and $R\left(s_{n},\;a_{n}\right)$ denotes the reward at step *n*. The value function *Q*^π^(*s_t_*, *a_t_*) measures the value of taking action *a_t_* on state *s_t_*. *V*^π^(*s_t_*) is the value of being at state *s_t_*. *V*^π^(s_t_) can be seen as a part of *Q*^π^(*s_t_*, *a_t_*). The remaining part of *Q*^π^(*s_t_*, *a_t_*) can be defined as the so-called advantage function A^π^(*s_t_*, *a_t_*) [43] as follows:(3)$$A^{\pi \;}\left(s_{t},a_{t}\;\right)=Q^{\pi \;}\left(s_{t},a_{t}\;\right)-V^{\pi \;}\left(s_{t}\right)\;\;\;$$

Intuitively, the advantage value shows how advantageous selecting the action is relative to the others at the same given state. Equation (2) can be rewritten according to Equation (4):(4)$$V^{\pi \;}\left(s_{t}\right)=E_{a_{t}∼\pi \left(s_{t}\right)}\left[V^{\pi \;}\left(s_{t}\right)+A^{\pi \;}\left(s_{t},a_{t}\;\right)\right]=V^{\pi \;}\left(s_{t}\right)+E_{a_{t}∼\pi \left(s_{t}\right)}\left[\;A^{\pi \;}\left(s_{t},a_{t}\;\right)\right]$$

Obviously, $E_{a_{t}∼\pi \left(s_{t}\right)}\left[\;A^{\pi \;}\left(s_{t},a_{t}\;\right)\right]=0$. To avoid the issue of identifiability, we subtracted the mean value from the prediction, and the Q-function of the dueling DQN can be defined as follows:(5)$$Q^{\pi \;}\left(s_{t},a_{t};\;\Theta ,\;\alpha ,\;\beta \;\right)=V^{\pi \;}\left(s_{t};\;\Theta ,\;\;\beta \right)+(A^{\pi \;}\left(s_{t},a_{t};\;\Theta ,\;\alpha \;\right)-\frac{1}{\left|\mathrm{Ac}\right|}\sum_{a_{t}^{\prime }}A^{\pi \;}\left(s_{t},a_{t}^{\prime };\;\Theta ,\;\alpha )\right)$$

Note that Θ, α, and β come from the dueling Q-network, as shown in Figure 8. Moreover, |Ac| is the size of the action space and $a_{t}^{\prime }\epsilon \mathrm{Ac}$. To make our RL agent more robust for more stable learning and to handle the problem of the overestimation of Q-values, the double Q-network [46] and fixed Q-targets [47] were also incorporated:(6)$$TD=Q^{\pi \;}\left(s_{t},a_{t};\;\Theta ,\;\alpha ,\;\beta \;\right)-\left[R\left(s_{t},\;a_{t}\right)+\gamma \cdot Q_{tar}^{\pi }\left(s_{t+1},argmax_{a_{t+1}}Q^{\pi \;}\left(s_{t+1},a_{t+1};\;\Theta ,\;\alpha ,\;\beta \;\right);\;\Theta {}^{\prime },\;\alpha {}^{\prime },\;\beta {}^{\prime }\;\right)\right]$$
where *TD* is the temporal difference; and $Q_{tar}^{\pi }$ is another dueling DQN network, as the target network and its parameters ($\Theta {}^{\prime },\;\alpha {}^{\prime },\;\beta {}^{\prime }$) were fixed and copied from the dueling DQN $Q^{\pi \;}$every *m* step (*m* = 20). To update the parameters (Θ, α, β) from the dueling DQN as shown in Figure 7, we trained our RL agent by minimizing the loss function:(7)$$l\left(\Theta ,\;\alpha ,\;\beta \right)=E\left[f_{l}\left(TD\right)\right]$$
where *E* is the expectation. As the disadvantage of the L2 loss is the tendency to be dominated by outliers, we used the Huber loss as the loss function $f_{l}$:(8)$$f_{l}\left(x\right)=\left\{\begin{array}{c} \left|x\right|-0.5\;\;\;\;\;if\left|x\right|\geq 1\\ 0.5*x^{2}\;\;\;\;\;\;\;if\left|x\right|<1 \end{array}\right.$$

### 4.4. Prioritized Experience Replay

Prioritized experience replays [48] is a technique to enable the RL agent to remember and reuse experiences from the past and to replay important transitions more frequently. Prioritized experience replay is highly useful for replaying some less frequent experiences. Here, we used the “Prioritized Replay Buffer” code from Open AI Gym (version 0.15.4) [49]. Finally, our RL agent was trained in the double-dueling deep Q-learning network with fixed q-targets and prioritized experience replay.

### 4.5. Fragment Library Design

The fragment-based approach to drug discovery (FBDD) has been established as an efficient tool in searching for new drugs [50]. The idea of FBDD is that proper optimization of each unique interaction in the binding site and subsequent incorporation into a single molecular entity should produce a compound with a binding affinity that is the sum of the individual interactions. However, the widely used fragment libraries consider only the diversity of fragments, such as the ZINC fragment database. Therefore, there is a very low probability of achieving the desired bioactivity for a given protein.

To combine FBDD with our RL framework, we first collected and built a SARS-CoV-2 3CL^pro^ inhibitor dataset containing 284 reported molecules. We adopted the improved BRICS algorithm [46] to split these molecules to obtain the fragment library target on SARS-CoV-2 3CL^pro^, as demonstrated in the flowchart in Figure 1 (yellow box). An elaborate filtering cascade is accompanied by manual inspection, and the rules can be changed based on the needs of different studies. Our fragment library contained 316 fragments with molecular weights of <200 Da and minimum and maximum numbers of non-hydrogen atoms of 1 and 25, respectively. The fragments that come directly from existing inhibitors based on the chemical retrosynthetic rules are always true substructures and may have high bioactivity in targeting 3CL^pro^. The quality of the designed fragment library can directly affect the properties of the chemical environment of the ADQN–FBDD.

### 4.6. Core Selection

Previous studies have identified various scaffolds or core structures with advantageous characteristics in terms of the activity for a particular target [51,52,53]. Core structure selection is the starting point in scaffold-based drug discovery. However, choosing or designing a proper initial scaffold is not a simple task, and medicinal chemists may require considerable experience. Nevertheless, there are several reported core structures targeting SARS M^pro^ [54]. We chose 4-aminopent-2-enal and 3-amino-2-oxobutanal as the starting cores (Figure 9) because both cores have been validated to generate covalent bonds with the Cys145 of SARS or SARS-CoV-2 3CL^pro^.

### 4.7. Reward Design

Most reported RL methods use the complete structural information of a positive drug or inhibitor as the template [38,44]. They design a reward function for the RL agent to learn to regenerate the template structure or generate highly similar structures to the template. This approach may be useful in testing the performance of RL methods but may not be suitable in real-world drug design because the complete structural information of the novel molecule is unknown. A more practical approach is to learn the structural features of existing drugs or inhibitors in the focused chemical space for a specific protein target. Instead of focusing only on the diversity of molecular structures, we explored the possibility of generating novel molecules based on the existing knowledge. Here, we designed a deep reinforcement learning reward (DRL-reward) function consisting of the final property score with the specific fragments (CSF) score and pharmacophore score as follows:(9)$$R\left(s\right)=w_{pro}\cdot f_{pro}\left(s\right)+w_{con}\cdot f_{con}\left(s\right)+w_{pha}\cdot f_{pha}\left(s\right)$$
(10)$$f_{pro}\left(s\right)=\left\{\begin{array}{c} 1\;\;if\;QED\left(S\right)>0.1\;\\ 0\;\;else\;\;\;\;\;\;\;\;\;\;\;\;\;\;\;\;\;\;\;\;\;\;\;\;\;\;\; \end{array}\right.$$
where $w_{pro}$ represents the weight for the quantitative estimate of drug-likeness (*QED*), and its default value is 0.1; and $f_{pro}$ represents *QED*. *QED* values can range from 0 (all properties are unfavorable) to 1 (all properties are favorable), calculated based on eight molecular properties [55]. The score function $f_{csf}\;$of *CSF* is as follows:(11)$$f_{CSF}\left(s\right)=\left\{\begin{array}{c} F_{csf}\left(s\right)\;\;\;\;\;\;if\;F_{CSF}\left(s\right)>0.9\;\;\;\;\;\;\;\;\;\;\;\;\;\;\;\;\;\;\;\;\;\;\;\;\;\;\;\\ \;\;\;\;\;\;\;\;\;\;\;\;\;\;\;\;\;\;\;\;\;\;\;\;\;\;\;0\;\;\;\;\;\;\;\;\;\;\;\;\;\;\;else\;\;\;\;\;\;\;\;\;\;\;\;\;\;\;\;\;\;\;\;\;\;\;\;\;\;\;\;\;\;\;\;\;\;\;\;\;\;\;\;\;\;\;\;\;\;\;\;\;\;\;\;\;\;\;\;\;\;\;\;\;\;\;\;\;\;\;\;\;\;\;\;\;\;\; \end{array}\right.$$
(12)$$F_{CSF}\left(s\right)=\frac{n_{match}}{N_{total\;}}$$

The binding site of SARS-CoV-2 3CL^pro^ (Figure 10) is commonly divided into the catalytic activity center (His41 and Cys145; S1′) and several subsites, defined as S1 (His163, Glu166, Phe140, Leu141, and Asn142), S1′ (His41, Cys145, Gly143, and Ser144), S2 (Tyr54, Asp187, His41, Arg188, His164, and Met49), and S3 (Thr190, Gln192, Glu166, Met165, Leu167, and Gln189). Each subsite may have its favorable binding fragment. When generated structures (including intermediates) contain these favorable fragments, $f_{csf}$ is equivalent to giving an additional reward to our RL agent. Moreover, $w_{con}$ controls the contribution of the biased fragment to the reward signal, and the default value is 0.6; $n_{match}$ represents the number of biased fragments matched in one generated structure; $N_{total\;}$is the number of biased fragments defined based on our knowledge from related work; and $f_{pha}$ represents the score function of the pharmacophore, which mainly depends on the ligand–protein interaction mode from the crystal structure (PDB ID: 6LU7):(13)$$f_{pha}\left(s\right)=\left\{\begin{array}{l} 1\;\;\;\;\;if\;matchs\;the\;defined\;pharmacophores\;\;\;\;\;\;\;\;\;\;\;\;\\ 0\;\;\;\;\;else\;\;\;\;\;\;\;\;\;\;\;\;\;\;\;\;\;\;\;\;\;\;\;\;\;\;\;\;\;\;\;\;\;\;\;\;\;\;\;\;\;\;\;\;\;\;\;\;\;\;\;\;\;\;\;\;\;\;\;\;\;\;\;\;\;\;\;\;\;\;\;\;\;\;\;\;\;\;\;\;\;\;\;\;\;\; \end{array}\right.$$

The pharmacophore plot is shown in Appendix A. Moreover, $w_{pha}$ controls the contribution of the pharmacophore score to the reward, and the default value is 0.4.

### 4.8. Molecule Generation and Selection

As discussed in Section 4.7, addressing the reward design, our reward function considered the molecular descriptor threshold (QED > 0.1), the defined pharmacophore mode, and biased fragments. A total of 4922 unique valid structures were automatically generated, and all matched the defined rules, using the ADQN–FBDD without any pre-training, as required by other methods [30,36,45,56,57]. Next, all molecules with a high deep-reinforcement-learning score (DRL score: R(S) >= 0.6) were selected (47 molecules), as they were more drug-like, contained more favorable fragments of SARS-CoV-2 3CL^pro^, and covered more pharmacophore modes compared with other candidates obtained from multiple learning steps. These 47 unique molecules were prepared to generate at least one conformation with local energy minimization, using the OPLS-2005 force field of the “ligand prepare” module of Schrödinger 2015 software. The 47 unique molecules generated a total of 163 3D conformations before docking into the substrate-binding site of SARS-CoV-2 3CL^pro^. Considering the balance between precision and calculation time, the stand-precision (SP) Glide [58] was initially used to predict the possible non-covalent binding poses in this binding site and the binding site grid centered on the original ligand N3 [59] with 20 Å buffer dimensions. Following non-covalent docking, we also determined the covalent docking poses and scores for the 47 molecules. We reordered the docking results mainly based on the covalent docking score and the RMSD difference between the covalent and non-covalent poses (Appendix A).

## Figures and Tables

**Figure 1 biomolecules-12-00746-f001:**
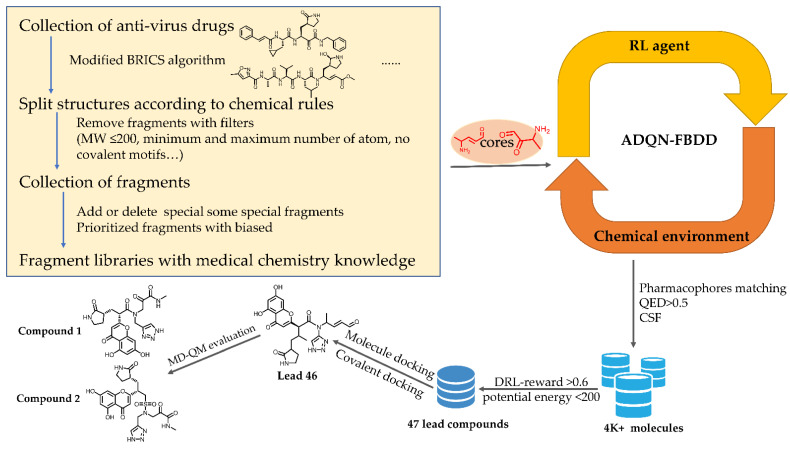
Art for SARS-CoV-2 3CL^pro^ lead compound generation.

**Figure 2 biomolecules-12-00746-f002:**
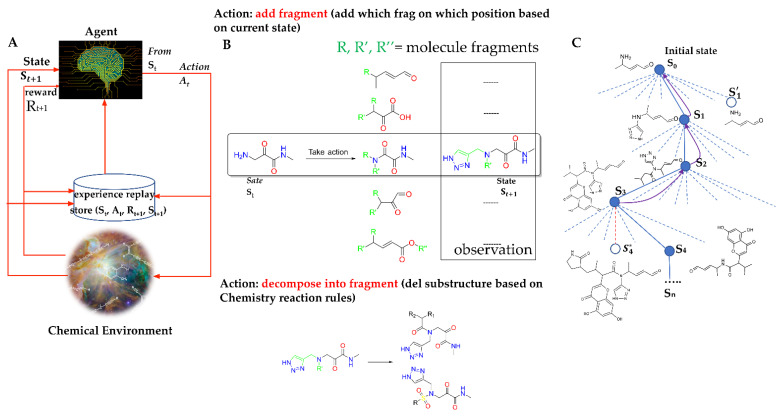
(**A**) Framework of the ADQN–FBDD. (**A**) The ADQN–FBDD consists of a reinforced agent, a prioritized experience replay algorithm, and a chemical environment to perform chemical structure generation. The agent selects an action (insertion, deletion, or none) for the intermediate molecular fragment at each step to generate a new molecule that can maximize the cumulative rewards. The prioritized experience replay algorithm allows the agent to repeat the molecule generation based on the updated maximization of rewards. The chemical environment assesses the agent’s actions according to the predefined chemical rules and provides rewards. (**B**) Example of fragment-based actions. (**C**) The solid lines represent taken actions, including the addition or deletion of different fragments during an episode. The dashed lines represent actions that the RL agent was considered but did not take. An exploratory action is represented by the red dashed line, which was taken even though another sibling action, the one leading to S*, was ranked higher. The exploratory action did not result in any learning; however, other actions did, resulting in updates as demonstrated by the curved arrows where estimated values moved up the tree from later nodes to earlier nodes.

**Figure 3 biomolecules-12-00746-f003:**
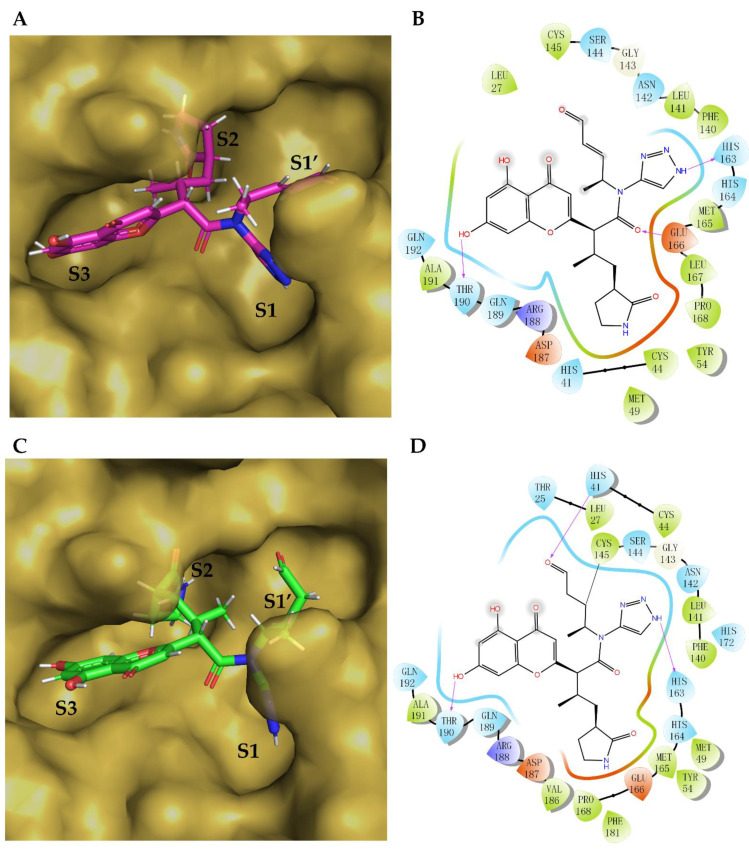
Lead compound #**46** generated by AI. (**A**) Non-covalent docking model of SARS-CoV-2 3CL^pro^ (brown surface) with the bound lead compound #**46** (magenta sticks). The triazole ring binds to the S1 subsite of the active catalytic center, the covalent fragment of the α, β-unsaturated aldehyde binds to the S1′ subsite, the β-lactam ring binds to the S2 subsite, and 5,7-dihydroxy chromone binds to the S3 subsite. (**B**) Two-dimensional (2D) view of the non-bonding interactions of lead compound #**46** in complex with 3CL protease based on non-covalent docking. The triazole ring, ketoamide group, and phenolic hydroxyl group form hydrogen bonds (H-bonds) with His163, Glu166, and Thr190, respectively. (**C**) Covalent docking model of compound #**46** (green sticks) with 3CL protease (brown surface), similar to the non-covalent docking model. (**D**) Two-dimensional view of ligand interactions between compound #**46** and protease under covalent docking. The triazole ring forms an H-bond with His163. The α, β-unsaturated aldehyde forms a covalent bond with Cys145, i.e., the key residue in the catalytic center of the protease, resulting in covalent inhibition. The aldehyde carbonyl group forms an H-bond with His41. The hydroxyl group of chromone at position 7 forms an H-bond with Thr190.

**Figure 4 biomolecules-12-00746-f004:**
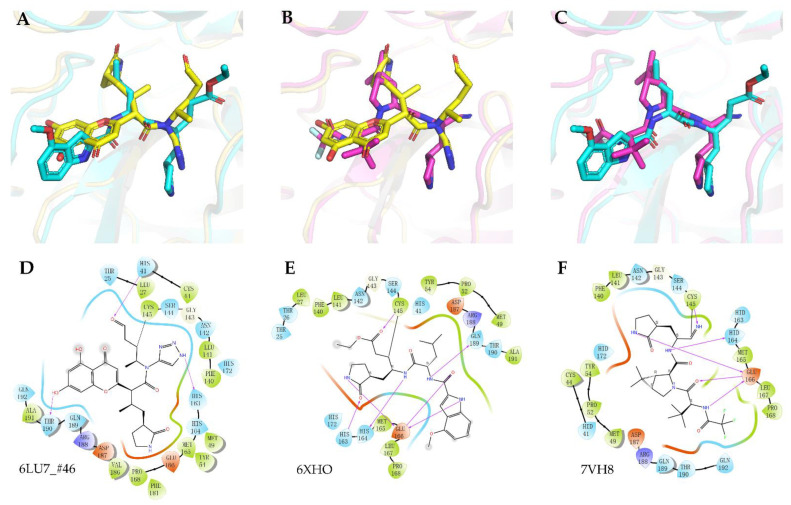
Co-crystal structure of the covalent adduct of (**A**) 6XHO (cyan) and (**B**) 7VH8 (magenta) bound to SARS-CoV-2 3CL^pro^ together with the alignment of (**C**) lead compound #**46** (yellow), and the detailed interactions of the complex: (**D**) 6LU7_#**46**, (**E**) 6XHO, and (**F**) 7VH8.

**Figure 5 biomolecules-12-00746-f005:**
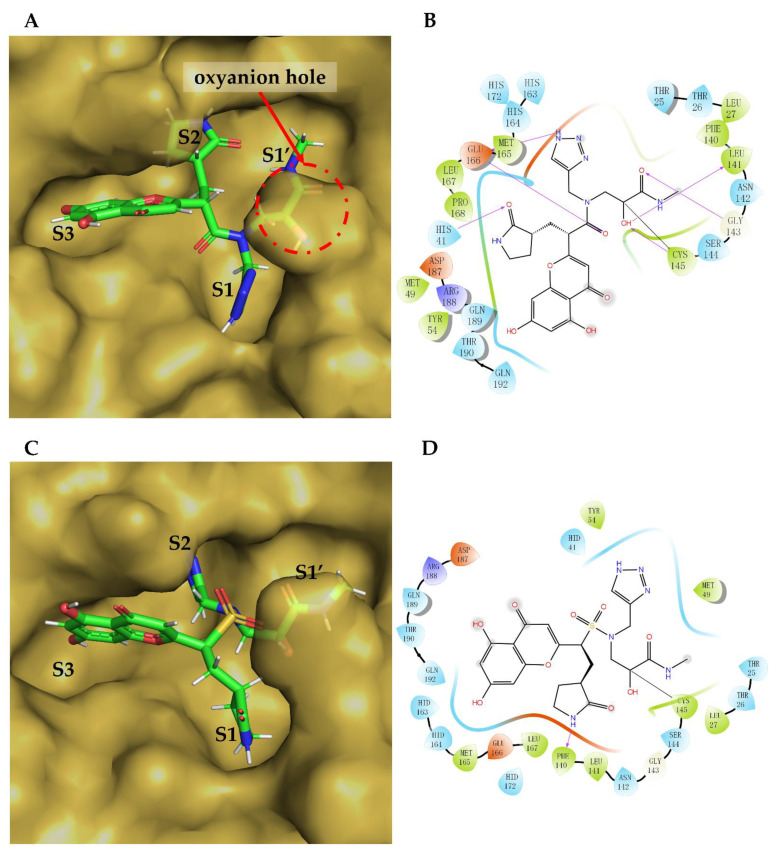
Covalent binding models of compounds **46–14–1** and **46–14–2** in complex with SARS-CoV-2 3CL^pro^. (**A**) Covalent docking model of compound **46–14–1** (green sticks) with 3CL protease (yellow-orange surface). The oxyanion hole formed by the segment of α-ketoamide is shown in the red circle. (**B**) Detailed view of the interactions between compound **46–14–1** and 3CL^pro^. (**C**) Covalent docking model of compound **46–14–2** with 3CL protease. Molecule **46–14–2** is shown as green sticks, and the protein is shown as a brown surface. (**D**) Two-dimensional view of the interactions between compound **46–14–1** and 3CL^pro^.

**Figure 6 biomolecules-12-00746-f006:**
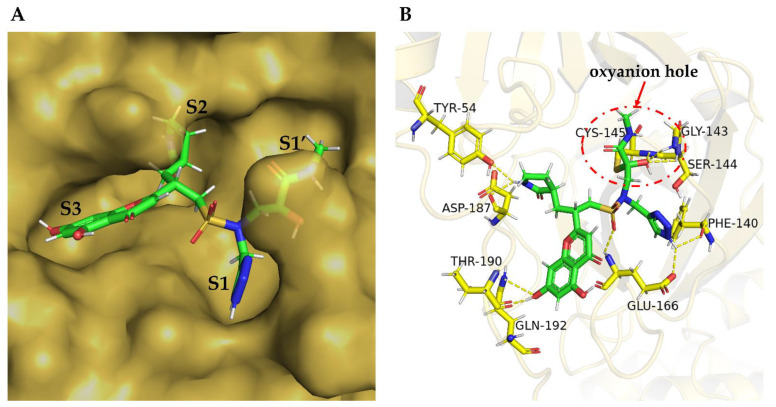
Covalent binding model of compound **46–14–3** in SARS-CoV-2 3CL^pro^. (**A**) Surface representation of SARS-CoV-2 3CL^pro^ (brown) complexed with **46–14–3** (green sticks). (**B**) Stereoscopic view of **43–14–3** in the substrate-binding pocket of SARS-CoV-2 3CL^pro^ at 4 Å. Molecule **43–14–3** is shown as green sticks. Residues forming H-bonds are shown as yellow sticks. Yellow dashes represent the H-bonds, and the oxyanion hole is in the red circle.

**Figure 7 biomolecules-12-00746-f007:**
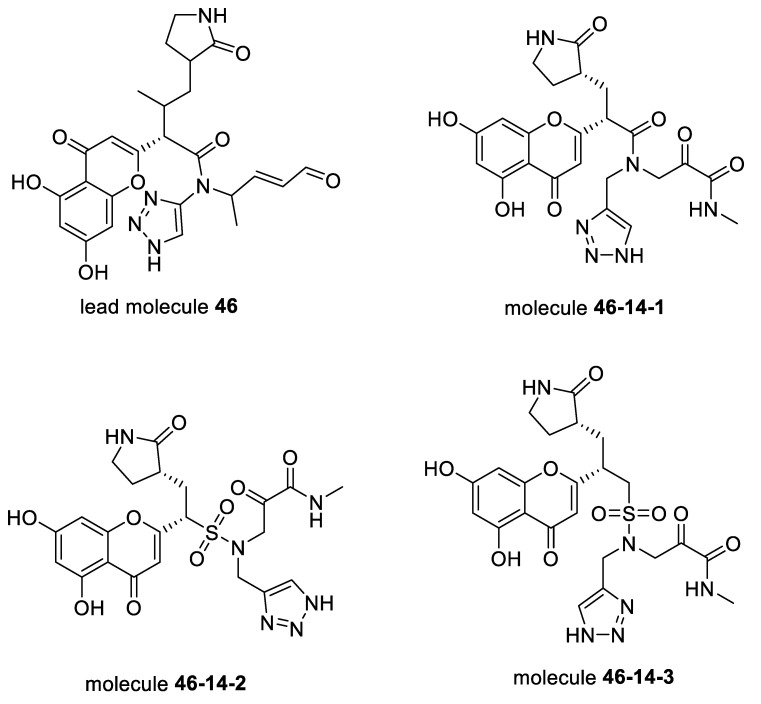
Structures of optimized compounds.

**Figure 8 biomolecules-12-00746-f008:**
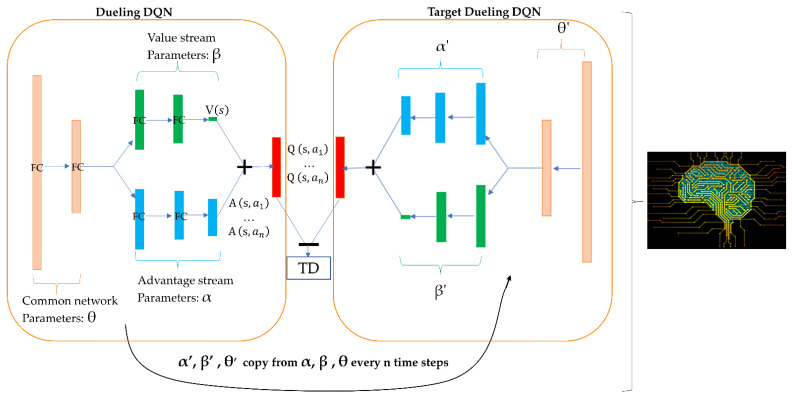
Architecture of the applied deep Q-learning network (ADQN). TD is the temporal difference error.

**Figure 9 biomolecules-12-00746-f009:**
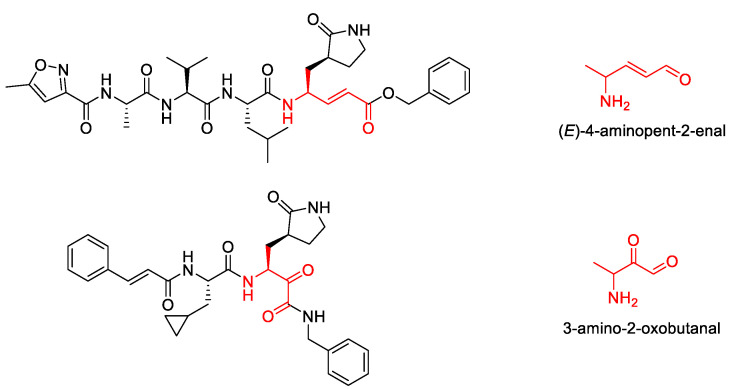
Structure of the chosen cores.

**Figure 10 biomolecules-12-00746-f010:**
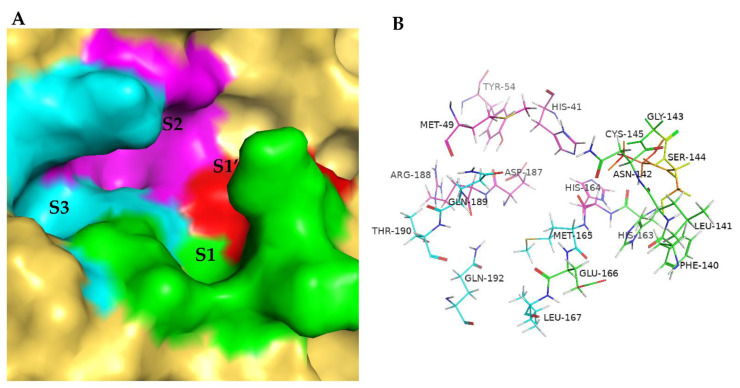
Binding site of SARS-CoV-2 3CL^pro^. (**A**) Surfaces of subsites that complement the substrate-binding pocket that are labeled as S1 (green), S1′ (red), S2 (magenta), and S3 (cyan). (**B**) Key residues of the binding site that are presented as green, red, magenta, and cyan lines (S1, S1′, S2, and S3). Images of the binding site were generated by using PyMol (http://www.pymol.org/).

**Table 1 biomolecules-12-00746-t001:** Comparisons of existing deep-learning-based molecule design tools.

Representative Tool	Method	Training/Performance
RNNs [35]	Recurrent neural networks	Extensive training data are required to learn the context of the atom composition; may generate many invalid SMILES
GENT*RL* [36]	Variational-autoencoder-based models	Large training data are required to learn the atom distribution from training SMILES; may still produce some invalid SMILES
ORGAN [37],ORGANIC [38]	Generative adversarial networks	Use adversarial attacks to enhance the success rate; however, this still cannot guarantee valid SMILES
MolDQN [39]	Reinforcement-learning-based models	Employ self-learning without training data and all generated SMILES are valid

## Data Availability

The code for the ADQN–FBDD and related data in this paper will be available at https://github.com/tbwxmu/2019-nCov [40].

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
