# Peer review of "AI-Aided Design of Novel Targeted Covalent Inhibitors against SARS-CoV-2"

_biomolecules, 2022, doi:10.3390/biom12060746_

Round 1

Reviewer 1 Report

In this article, authors have done a nice work in expanding the use of AI for drug design with SARS-Cov2 as the focus. The methods are well explained in details and the flow of the paper makes sense. However there are few points which can improve the quality of the paper.

  1. Figure 1,2 should be magnified as its very difficult to see anything now.
  2. For figure 3B and D, the 2D representation is very nice however they should be made more clearer or better resolution. Its difficult to read the residues and the interactions. Same advice for similar figures later on.
  3. I am really curious about the differences between covalent and non-covalent docking models. based on the text for 3B and 3D, the H-bonding is retained, although looking at the figures it seems for covalent docking the trizole ring has moved away from the His163. the authors should also talk about the differences between the models and not only similarities. also how does the algorithm changes for these two models as the aldehyde groups which makes a covalent bond with Cys145 of Mpro remains in both the docking models. because what is the point of making two docking models, if under experimental conditions both will form a covalent bond.
  4. Figure 4 needs more labeling or clarification. Although authors state in line 194 "compared their interactions", there is no comparison as there are no residues labeled to see the differences/similarities. I would recommend to replace this whole figure with 2D representation as done before and compare the interactions between solved structures (6XHO and 7VH8) and the lead molecule 46. This will be more informative as readers can see both the differences and similarities.
  5. Authors have mostly spoken about H-bonding interactions as the criteria for improving the molecule. What about hydrophobic interactions?
  6. there is a typo in line 226. It should be 5C and D.
  7. For line 250, it should be 46-14-2.
  8. there are two figures labeled as figure 7. Please correct the numbering for subsequent figures. In total there are 10 figures and not 9.

Author Response

In this article, authors have done a nice work in expanding the use of AI for drug design with SARS-Cov2 as the focus. The methods are well explained in details and the flow of the paper makes sense. However there are few points which can improve the quality of the paper.

  1. Figure 1,2 should be magnified as its very difficult to see anything now.

Answer: Thanks for this suggestion. We have improved those figures resolution. It should be easy to see now.

For figure 3B and D, the 2D representation is very nice however they should be made more clearer or better resolution. Its difficult to read the residues and the interactions. Same advice for similar figures later on.

Answer: We apologize for the quality of the figures. We have improved the resolution of all the figures in the revised manuscript.

  1. I am really curious about the differences between covalent and non-covalent docking models. based on the text for 3B and 3D, the H-bonding is retained, although looking at the figures it seems for covalent docking the trizole ring has moved away from the His163. the authors should also talk about the differences between the models and not only similarities. also how does the algorithm changes for these two models as the aldehyde groups which makes a covalent bond with Cys145 of Mpro remains in both the docking models. because what is the point of making two docking models, if under experimental conditions both will form a covalent bond.

Answer: Thanks for your interest. The main differences from non-covalent to covalent docking models are mainly presented by the 3D pose, as we mentioned in the original text ‘a good covalent docking score and a small RMSD value (docking affinity: -8.722 kcal/mol, KabschRmsd 1.71 Å) in Table S1.’ According to the 3D poses, the difference may be mainly from the moving of aldehyde groups and γ-lactam instead of the triazole rings moving (considering the 2D is a projection of 3D, which cannot reflect the relative distance in 3D space). The algorithms for docking and covalent docking were also different. Briefly, covalent docking algorithms allow the ligand and side chain of CYS145 to be flexible and movable; all rest parts of the protein are rigid. The non-covalent docking only allows the ligand to be flexible to change. A detailed introduction to both algorithms can be found in the paper "Docking covalent inhibitors: A parameter-free approach to pose prediction and scoring." Aldehyde group can form a covalent bond with CYS145, but considering the toxicity of aldehyde, which readily forms a covalent bond with other proteins or peptides CYS, e.g., GSH. We are not interested in the simple aldehyde group. #46 has “O=C-C=C” as a Michael acceptor, which could form a covalent bond with CYS145 with the Michael addition. For the formed covalent bond, we assume the reaction group of the ligand should keep a favorable distance to the SH atom from CYS145 to compose an “initial non-covalent complex” as we mentioned in the Introduction section. We calculated the difference from “final covalent complex” to “initial non-covalent complex” to evaluate those covalent inhibitors. A ligand’s difference (the RMSD value) from non-covalent to covalent binding is small, meaning it doesn’t need to change its 3D pose too much to form the covalent bond, which is why we used two docking models (non-covalent and covalent).

  1. Figure 4 needs more labeling or clarification. Although authors state in line 194 "compared their interactions", there is no comparison as there are no residues labeled to see the differences/similarities. I would recommend to replace this whole figure with 2D representation as done before and compare the interactions between solved structures (6XHO and 7VH8) and the lead molecule 46. This will be more informative as readers can see both the differences and similarities.

Answer: We have also added the 2D representation to compare the interactions between the lead molecule 46 and solved structures (6XHO and 7VH8). However, considering 2D cannot fully display the pose of inhibitors, we still keep the original 3D figures to highlight the 3D alignment indicating the similarity of orientation.

  1. Authors have mostly spoken about H-bonding interactions as the criteria for improving the molecule. What about hydrophobic interactions?

Answer: Thank you for pointing this out. The characteristics of this 3CLpro have determined that the main driving force for the binding inhibitors to proteins is hydrogen bond interaction when talking about non-covalent interaction. Of course, the hydrophobic interactions are also significant but not the dominant factor in this case. As mentioned in the 2D pictures, the green residues and bands represent the hydrophobic interaction between the residues and inhibitors.  #46’s H-bond is not strong enough compared with Pfizer’s two drugs. It indicated that #46 should be optimized and 46-14-3 has an equivalent number of H-bonds with Pfizer’s two drugs.

  1. there is a typo in line 226. It should be 5C and D.

Answer: Good catch! We have corrected this sentence.

  1. For line 250, it should be 46-14-2.

Answer: Thanks for your careful review. It is corrected in the revision.

  1. there are two figures labeled as figure 7. Please correct the numbering for subsequent figures. In total there are 10 figures and not 9.

Answer: Thanks for this point. We have corrected it.

Reviewer 2 Report

In this manuscript, Tang et al describe a deep reinforcement learning approach to designing novel molecular structures that target SARS-CoV-2. Thereby, it makes use of the fragments of the 284 known SARS-2CLpro inhibitors, thus increasing the possibility for the proposed structures to act efficiently against SARS-CoV-2. With the suggested method, the authors designed 47 candidate molecules, selected out of those the most promising one, and optimized its structure in three distinct ways. The optimized compounds are currently subject to experimental validation.

Of course, the experimental results would complement the paper in a very strong way. Yet the approach is of interest on its own, and the authors rightfully suggest that the obtained in silico results may serve as a basis for further investigations by other specialists as well.  Therefore, in my opinion, this paper can be published in Biomolecules.

It might make sense, however, to comment in more detail on the choice of the threshold values used by the method (those specified in the Methods section).

Typos:

Line 226: Figure 4C and D --> Figure 5C and D.

Line 399: >1 and <= 25 --> 1 and 25

Formula (11): “0 else” in the first line on the right-hand side is redundant.

Author Response

In this manuscript, Tang et al describe a deep reinforcement learning approach to designing novel molecular structures that target SARS-CoV-2. Thereby, it makes use of the fragments of the 284 known SARS-2CLpro inhibitors, thus increasing the possibility for the proposed structures to act efficiently against SARS-CoV-2. With the suggested method, the authors designed 47 candidate molecules, selected out of those the most promising one, and optimized its structure in three distinct ways. The optimized compounds are currently subject to experimental validation.

Of course, the experimental results would complement the paper in a very strong way. Yet the approach is of interest on its own, and the authors rightfully suggest that the obtained in silico results may serve as a basis for further investigations by other specialists as well.  Therefore, in my opinion, this paper can be published in Biomolecules.

It might make sense, however, to comment in more detail on the choice of the threshold values used by the method (those specified in the Methods section).

Answer: Thanks for your suggestion. Those thresholds were selected based on the DQN study from the paper “Human-level control through deep reinforcement learning”. We have included this explanation in our revised version (lines531-532).

Typos:

Line 226: Figure 4C and D --> Figure 5C and D. 

Answer: We have corrected this line in the revision.

Line 399: >1 and <= 25 --> 1 and 25

Answer: Thanks for pointing it out. We have fixed it.

Formula (11): “0 else” in the first line on the right-hand side is redundant. 

Answer: We have deleted it from Formula (11).

Reviewer 3 Report

The article "AI-aided design of novel targeted covalent inhibitors against SARS-CoV-2" is well written and methodologically well structured. Although the development of this type of method is not new, its application is attractive for developing new structural analogues with potential biological activity.

The article seems to be correct for the special issue of the journal. However, I have minor suggestions for the authors.

1° Correct the superscripts of the protein names (¿3CLpro or 3CLpro?).

2° To deepen in the discussion the methodological advantages and projections of the use of fragment-based drug design.

Author Response

The article "AI-aided design of novel targeted covalent inhibitors against SARS-CoV-2" is well written and methodologically well structured. Although the development of this type of method is not new, its application is attractive for developing new structural analogues with potential biological activity.

The article seems to be correct for the special issue of the journal. However, I have minor suggestions for the authors.

 1° Correct the superscripts of the protein names (¿3CLpro or 3CLpro?).

Answer: Thanks for this point. We have corrected it.

2° To deepen in the discussion the methodological advantages and projections of the use of fragment-based drug design.

Answer: Thanks for this suggestion. We have added a new paragraph to discuss the methodological advantages and projections of the use of fragment-based drug design in the discussion part.

Round 2

Reviewer 1 Report

The authors have answered all my queries. I have no more queries. This is a good work.